# Development of Radiation-Tolerant HTS Magnet for Muon Production Solenoid

**Toru Ogitsu \***, **Masami Iio, Naritoshi Kawamura** and **Makoto Yoshida**

KEK, High Energy Accelerator Research Organization, 1-1 Oho, Tsukuba 305-0801, Japan; iio@post.kek.jp (M.I.); nari.kawamura@kek.jp (N.K.); makoto.yoshida@kek.jp (M.Y.)

**\*** Correspondence: toru.ogitsu@kek.jp

**Abstract:** Superconducting magnets are widely used in accelerator science applications. Muon production solenoids are applications that have recently attracted considerable public attention, after the approval of muon-related physics projects such as coherent muon to electron transition or muon-to-electron-conversion experiments. Based on its characteristics, muon production solenoids tend to be subjected to high radiation exposure, which results in a high heat load being applied to the solenoid magnet, thus limiting the superconducting magnet operation, especially for low-temperature superconductors such as niobium titanium alloy. However, the use of high-temperature superconductors may extend the operation capabilities owing to their functionality at higher temperatures. This study reviews the characteristics of high temperature superconductor magnets in high-radiation environments and their potential for application to muon production solenoids.

**Keywords:** ReBCO; muon production; radiation hardness

## 1. Introduction

A muon production solenoid magnet generates a magnetic field that captures charged secondary particles, such as pions or muons, which are produced by a production target. In order to achieve high-efficiency muon production, the solenoid should be placed near the target, resulting in high radiation exposure. The most extreme case is the production solenoids planned for muon rare decay experiments such as the coherent muon to electron transition (COMET) [1] experiment at the Japan Proton Accelerator Research Complex (J-PARC) Japan, or the muon-to-electron-conversion (Mu2e) experiment [2] at the Fermi National Accelerator Laboratory (FNAL) U.S.A. Both the production solenoids, which are wound from the low-temperature superconductors (LTS), instrument the production target inside its aperture, resulting in a large aperture with thick radiation shielding. In the case of the COMET production solenoid [3,4], the aperture is 1.3 m in diameter with a 44-cm-thick Tungsten hybrid shield. Even with this design, the power of the primary proton beam that hits the target is limited to 56 kW [5].

Production solenoids that have higher radiation exposure levels are generally designed with normal conducting magnet technologies. In the J-PARC Material and Life Science Experiment Facility (MLF), a normal conducting solenoid is used for the ultra-slow muon beam line [6]. The solenoid is located about 0.6 m from the 50-kW target (1 MW proton beam with a 5% dump) and produces a central field of about 0.3 T with an aperture diameter of about 0.5 m, and it consumes about 50 kW of electric power. It also requires the treatment of large amounts of radiation active materials produced in the cooling water.

In J-PARC MLF, a new production target with a 1-MW proton beam is now proposed, and the target (named TS2) will be utilized for both high-intensity cold neutrons and muons. For the muon

beam, the production solenoid will be placed in close proximity to the target, approximately 0.8 m, in order to achieve the highest beam intensity, which is 10 times higher than that of the original muon beam lines in the MLF. The solenoid may require a central field of the order of 1 T with an aperture diameter of approximately 0.6 m. Using normal conducting magnet technologies, the power consumption would be unreasonably large, while for LTS technologies, such as niobium titanium alloy (NbTi) superconductor, the required radiation shield thickness is too great to realize a realistic coil design that can produce the required magnetic field. With recent developments in the field of high-temperature superconductors (HTS), such as the rare-earth element barium copper oxide (ReBCO), where rare-earth (Re) can be gadolinium (Gd), europium (Eu), or yttrium (Y), various HTS conductors are now commercially available. For example the mass production technology of 10-mm-wide and 500-m-long GdBCO tape conductors with a current capacity of about 650 A in a 77-K self-field [7] is established. Various studies on HTS magnets with high-radiation environments as well as the radiation hardness measurements of various superconducting magnet materials, including HTS conductors have been performed. The review of these previous studies verifies the feasibility of the production solenoid using HTS technologies. The paper briefly summarizes the review results and provides an abstract design concept for production solenoids.

## 2. LTS Production Solenoid

As described in the introduction, production solenoids are being developed using LTS technologies. In this chapter, taking the COMET as an example case, the limitation of the LTS magnet in a high-radiation environment is discussed.

### 2.1. COMET Production Solenoid Design

The design of the COMET production solenoid is shown in Figure 1. The COMET production solenoid comprises a capture solenoid (CS, wound from two coils CS0 and CS1) and a matching solenoid (MS, wound from two coils MS1 and MS2). The CS generates a main capture solenoid field of approximately 5 T, while MS gradually reduces the field from the capture solenoid coils to the transport solenoid (TS1, wound from 6 coils TS1a, TS1b, TS1c, TS1d, TS1e, and TS1f). The main parameters of CS, MS, and TS1 are summarized in Table 1.

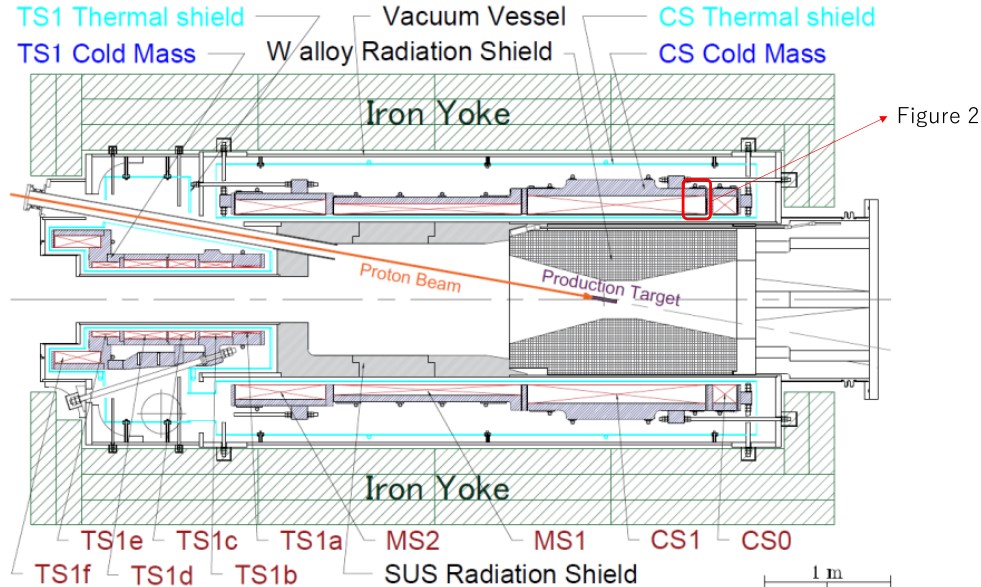

**Figure 1.** Design of the proposed COMET production solenoid.

**Table 1.** The main parameters of the COMET production solenoid coils.

| Coil Identification | Inner Diameter [mm] | Thickness [mm] | Length [mm] | No. of Turns | No. of Layer | Operation Current [A] |
|---|---|---|---|---|---|---|
| CS0 | 1344 | 152 | 180 | 35 | 9 | 2700 |
| CS1 | 1344 | 152 | 1391 | 270 | 9 | 2700 |
| MS1 | 1344 | 84 | 1468 | 285 | 5 | 2700 |
| MS2 | 1344 | 118 | 721 | 140 | 7 | 2700 |
| TS1a | 500 | 16 | 200 | 40 | 1 | 2700 |
| TS1b | 500 | 48 | 240 | 48 | 3 | 2581 |
| TS1c | 500 | 64 | 200 | 40 | 4 | 2700 |
| TS1d | 500 | 64 | 320 | 64 | 4 | 2619 |
| TS1e | 500 | 48 | 200 | 40 | 3 | 2538 |
| TS1f | 820 | 96 | 350 | 70 | 6 | 2916 |

*2.2. Thermal Design and Influence of Irradiation*

The thermal design of the CS1 shown in Figure 2 is based on the conduction cooling scheme that resembles that of detector solenoids for collider experiments [8,9]. The nine-layer 1.4-m long coil, which is wound from the aluminum-stabilized NbTi superconductor, is conductively cooled by 1-mm-thick pure aluminum strips that are installed between each coil layer. The strips were connected to cooling pipes that were cooled by forced-flow 2-phase helium. The connections are made at the coil ends, at which some of the strips are obstructed by the ramp and splice structures that are needed alternately at every other coil layer end. The structures of CS0, MS1, and MS2 are the same as those of CS1. An alternate design concept could include the use of cable in conduit technologies, which are widely used for fusion magnets. The design concept may allow higher heat deposit density; however, the overall cost required for the cable in the conduit design was extremely high, and was not acceptable. The other option is to provide a liquid helium casing around the coil and cool directly using liquid helium. This option was eliminated because the additional helium vessel increased the mass of the cold mass, which increased the overall heat load. This option also has the disadvantage of contamination of helium by any radioactive materials decomposed from the magnet assembly. In case of the conduction cooled magnet, because the helium is circulating in the cooling pipe made from aluminum, the helium contamination by the decomposed radioactive material is minimized. Tritium may be produced by thermal neutrons reacting to helium 3, which is naturally abundant about 1.4 ppm. Although the production of the tritium is limited by the minimized helium volume in the cold mass by choosing a pipe cooling scheme, some radiation safety measures are still required for maintenance and for the outlets of safety valves.

The greatest concern with radiation is the increase in the coil temperature, which is due to the heat load by the irradiation. The peak heat load in the coil is estimated to be approximately 0.035 W/kg. The temperature increase is enhanced owing to the thermal conductivity degradation on pure aluminum, which is due to the lattice defect introduced by hadronic irradiation. The degradation rate can be estimated to be approximately 0.03 n$\Omega$m for a neutron fluence of $10^{20}$ n/m$^2$, based on the measurement with reactor neutrons [10]. A model computation was performed using the full coil model with the nominal operation condition of the magnet, and the results indicate that the coil temperature gradually increased owing to the degradation of the thermal conductivity of the aluminum [11]. The coil temperature exceeds the current sharing temperature of 6.7 K, with an operation duration of approximately 90 days. Fortunately, the aluminum defect can be annealed 100% with the thermal cycle to room temperature, and the cooling performance can be fully recovered. However, the results indicate that for an LTS magnet with conduction cooling, the heat deposit density may be limited to the order of 0.035 W/kg.

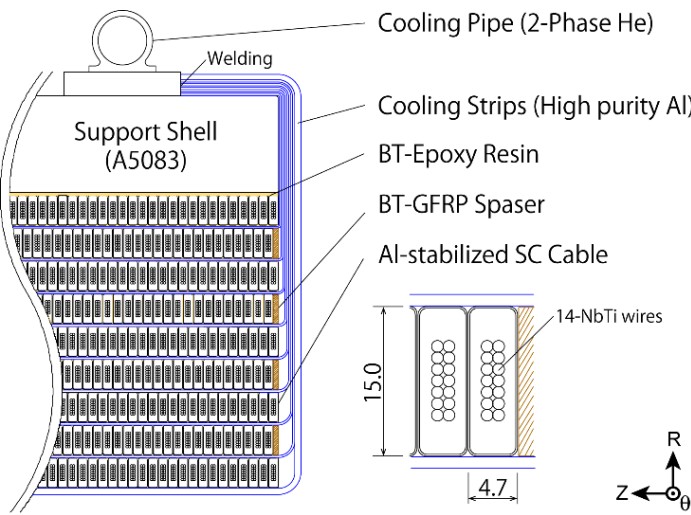

**Figure 2.** Details of thermal design of the capture solenoid coil (CS).

*2.3. Degradation of Thermal Conductivity by Neutron Irradiation on Pure Aluminum and Copper*

As discussed previously, during the design study phase of the COMET production solenoid, there was a concern related to the thermal conductivity degradation on pure aluminum. To verify the degradation of the aluminum used in the solenoid, measurements of the change in electric resistance with neutron irradiation at cryogenic temperatures were performed [10,12,13]. An example of the measured results for aluminum (RRR ~430) is summarized in Table 2.

**Table 2.** Summary of the resistance changes observed during the experiment.

| Period | Temperature | Neutron Fluence | Measured Resistance [1] |
|---|---|---|---|
| Before cool-down | 300 K | 0 | 1.3 mΩ |
| After cool-down | 10 K | 0 | 3.0 μΩ |
| During irradiation | 12–15 K | $1.4 \times 10^{15}$ n/m$^2$/s (45 h) | 3.1–5.7 μΩ Monotonic increase |
| After irradiation | 12 K | $2.3 \times 10^{20}$ n/m$^2$ | 5.6 μΩ |
| After warm-up to RT | 302 K | $2.3 \times 10^{20}$ n/m$^2$ | 1.3 mΩ |
| After the second cool-down | 12 K | $2.3 \times 10^{20}$ n/m$^2$ | 3.0 μΩ |

[1] The sample dimensions are 1 mm × 1 mm × 70 mm, and the voltage sensing length is ~45 mm.

During neutron irradiation, monotonic degradation of the electric resistance was observed, and the ratio of the degradation was approximately 0.027 nΩm for a $10^{20}$ n/m$^2$ fast neutron fluence ($E_n > 0.1$ MeV). Notably, in the case of aluminum, the thermal cycle to room temperature recovers the resistance to the original value completely. The measurements were performed for aluminum samples with different RRR values as well as some copper samples. In the case of aluminum, the degradation rates were approximately the same for all the samples, and complete recovery during the room temperature thermal cycle was always observed. For copper, the degradation rate was approximately three times lower than that of aluminum. However, recovery during the room temperature thermal cycle was insufficient, and additional damage was observed. It was also confirmed that residual degradation is accumulated by consecutive irradiation with thermal cycles [14].

## 3. HTS Magnet Studies under High-Radiation Environment Conditions

### 3.1. Studies of Conduction-Cooled Production Solenoid with HTS

While developing the COMET solenoid system, the team started to consider a more compact muon production solenoid that would lead to a compact muon source. A conceptual design model was developed based on the HTS conductor with conduction cooling using a cryo-cooler refrigerator [15]. The design of the HTS-based muon production solenoid is shown in Figure 3. The HTS coil, which has a 340-mm length, 340-mm inner diameter, and 390-mm outer diameter, is made from 34 double pancake coils wound from 4-mm-wide ReBCO tape. The coil nominal operation current is 105 A, resulting in a central field of approximately 3 T. The critical temperature of the coil is estimated to be higher than 50 K. The outer shell of the coil is directly cooled to 20 K by a cryocooler, which provides a refrigeration power of approximately 10 W at 20 K. A graphite muon production target, which is hit by an 8-GeV 3-kW proton beam, is placed in the coil aperture center with a 50-mm-thick tungsten alloy radiation shield.

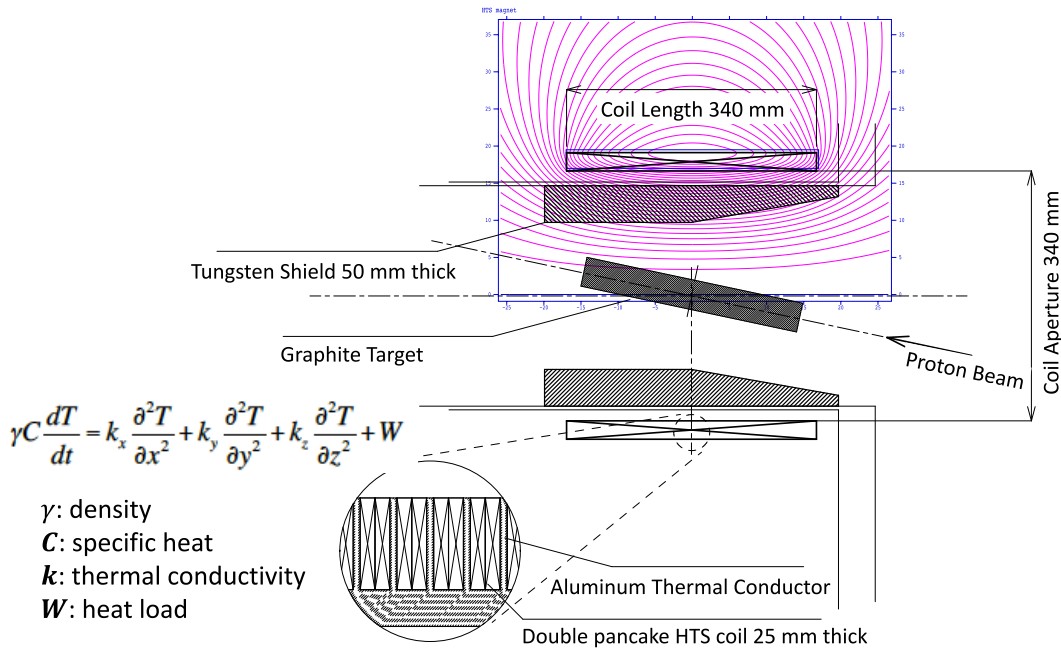

**Figure 3.** Conceptual design of HTS-based muon production solenoid.

Thermal computations were performed using numerical simulation code based on a thermal conduction model, as shown in Figure 3. The model is divided in each double pancake coil that is thermally connected to a 1.9-mm-thick aluminum thermal conductor with an RRR of 40 through the 25 μm polyimide insulation. Each pancake coil is assumed to have a radial thermal conductivity that is equivalent to a series of stainless steel, copper, and polyimide with a thickness ratio of 6:4:5, while for the longitudinal and azimuthal thermal conductivities, the thermal conductivity of stainless steel is used. For faster computation, all of the thermal conductivities were set to 18 K. The heat load distribution to the coil by the irradiation is computed using the Monte Carlo simulation code PHIT [16]. In the PHIT computation, the coil is considered to be a stainless steel cylinder with a 340-mm inner diameter and a 390-mm outer diameter. Based on the heat load computed by the PHITs, the temperature distribution in the coil is computed using the numerical simulation code. The maximum heat load is approximately 0.43 W/kg, while the temperature rise is only 1.6 K, which is far below the critical temperature of approximately 50 K. Comparing the case of the COMET production magnet in which the maximum heat load is approximately 0.035 W/kg, the HTS production can withstand more than

10 times the heat load with an excessive margin. This indicates that from a thermal design perspective, HTS magnets can be used in high-radiation environments that are unsuitable for LTS magnets.

It should be noted that the overall heat load to the coil is estimated to be approximately 7.1 W. The heat load can be cooled via a single cryo-cooler whose power consumption is approximately 7.5 kW.

### 3.2. Studies of FRIB Fragment Separator Magnets

Extensive studies have been performed on HTS magnets in high-irradiation environments by the National Superconducting Cyclotron Laboratory (NSCL) and the Brookhaven National Laboratory (BNL) for the Facility for Rare Isotope Beams (FRIB) fragment separator magnet system [17–19]. The major parameters of the HTS quadrupole magnet are summarized in Table 3 [17].

**Table 3.** The primary parameters of the HTS quadrupole magnet for FRIB.

| Parameter | Value |
|---|---|
| Pole Radius | 110 mm |
| Design Gradient | 15 T/m |
| Magnetic Length | 600 mm |
| Field Parallel to HTS Tape | ~1.9 T |
| Field Perpendicular to HTS Tape | ~1.6 T |
| Stored Energy | ~40 kJ |
| Operating Temperature | ~38 K (nominal) |
| Design Heat Load on HTS Coils | 5 kW/m$^3$ |

The quadrupole magnet is subjected to a large heat load and radiation loads because the magnet will be placed immediately after the target that produces various isotopes with a 400-kW 600-MeV proton beam or 200-MeV/amu ion beams. The HTS version is proposed as an option for the NbTi cable in conduit design developed at NSCL [20] to reduce the cryogenic operation cost by increasing the operation temperature to approximately 50 K instead of 4 K. In order to achieve high radiation tolerance, a 25-µm-thick stainless steel foil is placed between coil turns instead of organic films, such as a polyimide film, as a turn-to-turn insulator.

The model magnet comprises eight coils, of which four were wound from SuperPower 12-mm wide tape, while the other four were wound from 12-mm-wide ASC double tape. The coils were assembled using iron with a special support such that for the model magnet, iron was cooled with the coils (cold iron configuration). The coils were first tested individually at 77 K, then assembled in the cold iron configuration and tested with liquid nitrogen as well as gas helium at various temperatures.

### 3.3. Proton Irradiation Studies on YBCO Tapes

During the development of the FRIB magnet, an experimental program was launched to study radiation damage to the HTS tapes. Short samples of Yttrium-based ReBCO (i.e., YBCO) tapes obtained from SuperPower and ASC were irradiated with a 142-MeV proton beam at the Brookhaven Linear Isotope Production facility [18,21]. The critical current of the irradiated samples was first measured under a 77-K self-field condition, and the results demonstrated a monotonic degradation of the critical current [16]. The critical current measurements were then performed at a background field of 1 T for the irradiated samples. Although the behaviors of the two tapes varied significantly, neither showed any degradation or improvement of the critical current up to the irradiation level of 25 µA-h, indicating that the conductors can sustain the FRIB operation [21].

### 3.4. Neutron Irradiation Studies on ReBCO Tapes

Extensive studies on the influence of neutron irradiation on various gadolinum-based ReBCO (i.e., GdBCO) conductors have been performed by Michael Eisterer (TU-Wien) et al. [22–25]. A series

of irradiation tests were performed using the TRIGA MARK II research reactor at the Atominstitut. Samples of commercial GdBCO conductors from various suppliers were prepared and irradiated with various neutron fluences up to $4 \times 10^{22}$ m$^{-2}$, and were then measured to determine several superconducting properties such as the critical temperature or critical current. While the critical temperature appears to be degraded monotonically as a function of the fluence, the critical current measured at 30 K with a 15 T background field parallel to the c-axis of GdBCO crystal shows an improvement in the critical current in the fluence range below $2 \times 10^{22}$ m$^{-2}$ then starts to degrade. In most cases, critical currents start to degrade at lower fluence in the magnetic field parallel to the ab-plane than that to the c-axis; however, the values of the critical current remain higher with H parallel to the ab-plane.

Particularly, the spectrum of the irradiated neutron has two peaks: one at a low energy below 0.1 MeV, and the other at a high energy of approximately 1 MeV. The fraction of the fluence of low-energy neutrons (below 0.55 eV) is approximately 29%, while that of fast neutrons (above 0.1 MeV) is 36%. For the conductor that contains gadolinium in the HTS material, there is a significant influence from the low-energy neutron. The gadolinium contains two isotopes (Gd-155 and Gd-157, each having a natural abundance of ~15%) that have large cross sections for capturing thermal neutrons. The influence of thermal neutrons on GdBCO was compared using GdBCO samples with and without a cadmium foil wrapping to shield the thermal neutron. The results indicated that the unshielded sample demonstrated significant degradation of both the critical temperature and critical current with a neutron fluence of $0.6 \times 10^{22}$ m$^{-2}$, where the degradation is equivalent to that of $2.9 \times 10^{22}$ m$^{-2}$ irradiation of the shielded sample. The results mentioned previously are for the conductors that contain gadolinium, and were data obtained using a cadmium shield to avoid the influence of the thermal neutrons.

### 3.5. Radiation Hard Insulations

A major issue related to the magnet used in a high-radiation environment is the radiation hardness of the electric insulation material. For turn-to-turn insulation, most superconducting magnets use organic materials. Radiation hard organic materials, such as polyimide or cyanate-ester, have relatively high radiation hardness up to approximately 100 MGy [26], but from an engineering perspective, the order of 10 MGy may be the design limit. Another disadvantage of organic insulators is their low thermal conductivity that could significantly affect the conduction cooling magnet in a high-radiation environment because the material would be the major bottleneck of the heat transfer from the superconductor to the refrigerator. Inorganic insulation materials such as ceramics or glasses have considerably higher radiation hardness. In addition, these materials tend to have higher thermal conductivity.

For normal conducting magnet mineral insulation, hollow conductors are utilized for high radiation usage. The technology may be adjusted for superconductors, and one example is the NbTi cable in conduits, which was developed at NSCL [20]. Although it can achieve fully inorganic construction, there are obvious disadvantages with respect to the compaction factor, that is, current density. For conductors that require high-temperature heat treatment, such as niobium-tin compound superconductor ($Nb_3Sn$), ceramic insulation that is formulated during heat treatment at temperatures above 700 °C may be applicable. Reference [27] shows an example of the ceramic insulation applied to a $Nb_3Sn$ Rutherford cable. Thinner insulation with a titanium dioxide ($TiO_2$) sol-gel ceramic has also been developed for a bismuth strontium calcium copper oxide (Bi-2212) superconductor, which requires heat-treatment temperature above 800 °C [28].

For the ReBCO-coated conductor, such high-temperature heat treatment is not allowed. One way to overcome this issue is to apply ceramic insulation with high-temperature heat treatment to independent bases such as stainless steel tapes, and then to co-wind them with the HTS tape. This method was applied to the 32-T high field solenoid developed at the National High Magnetic Field Laboratory (NHMFL) [29]. A sol-gel silicon dioxide ($SiO^2$) ceramic layer was formed on the stainless steel tape at 300 °C for 10 s drying at 700 °C for 45 s. Although the technology is developed to sustain high stress

owing to its extremely high magnetic field, similar technology can be a favorable candidate for high radiation environments because of its high radiation hardness and high thermal conductivity.

Inorganic insulation using sol-gel technology with a heat treatment temperature of approximately 100 °C is also being developed [30]. A 40-µm-thick insulation layer of $SiO_2$ glass-like polymer with $Al_2O_3$ filler is formulated by heat treatment at about 100 °C, as shown in Figure 4. The reel-to-reel continuous coating process system, which is shown in Figure 5, and wet winding with inorganic adhesive is also being developed. Although it does not use the high-temperature heat treatment required for calcining, preliminary results from a thermal gravimetric analysis measurements reveal the high radiation hardness of the formulated insulation layer.

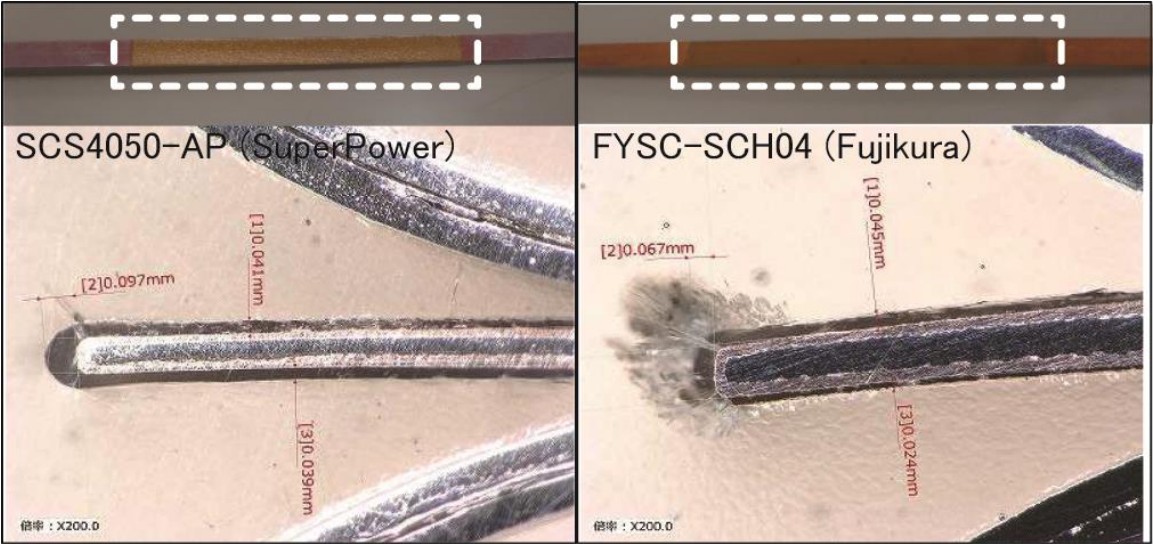

**Figure 4.** ReBCO tape short sample with inorganic insulation.

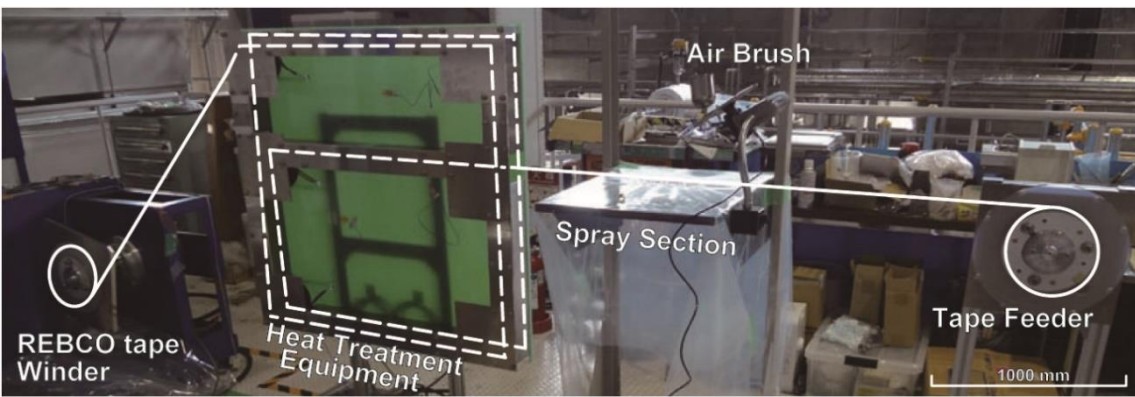

**Figure 5.** Reel-to-reel continuous coating system.

## 4. Muon Production Solenoid for Future High Intensity Muon Beam Line

Reviews of previous works thus far, have related to the development of HTS magnets for high radiation environments. This chapter presents a case study of a muon production solenoid magnet for a high-intensity muon beam line with an HTS conductor. In the MLF at J-PARC, the series of carbon and mercury targets that produce muon and neutron beams, respectively, are utilized as the first target station TS1. In the first target station, 10% of a 3-GeV 1-MW proton beam power is consumed at the first carbon target, while the rest of the power is dumped into the mercury target. To accommodate more users with higher intensity beams, the second target station TS2 is now proposed. For TS2, the 1-MW beam is fully dumped to a rotating tungsten target that produces both muon and neutron beams. In this

scenario, more than 10 times the intensity is expected for the muon beam line compared with that of TS1. In order to achieve the efficient capture of the pion produced by the target that decays to the muon while being transported in the muon transport line, the production should produce a solenoid field of approximately 1 T at a location as close as 1 m from the tungsten target. A normal conducting solenoid magnet is initially considered, but electric power consumption becomes unreasonably large, so the idea was not pursued.

In order to produce a magnetic field with reasonable power consumption, a superconducting solenoid option was considered. One of the proposed designs is shown in Figure 6, which shows the overall schematic view of the beam line system. The 3-GeV 1-MW proton beam is transported through the muon production solenoid and hits the Tungsten rotating target. Only backward-produced muons or pions are captured by the production solenoid, and are then transported to the muon transport line, where pions decay into muons. The design of the pion/muon production solenoid and transport solenoid is optimized to maximize the muon transport efficiency with the beam transport simulation code named G4beamline [31], which is based on Geant4 [32]. Simulated trajectories are shown in Figure 6. Pions drawn in dark blue lines are generated in the target position. The production solenoid captures pions, and they are tuned to be focused at the position of the first bending magnet. The first bending magnet selects the pion momentum which is adjusted to the intended backward-decay muon momentum. In the 8-m subsequent pion-to-muon decay solenoid magnet, pions decay into muons drawn in light blue lines. The beam line is also designed to transport surface muons (4 MeV) in a dedicated mode, where the transmission efficiency is maximized by using only the entrance and the exit part of transport solenoid drawn in red in Figure 6 under the principle proposed in the article [33].

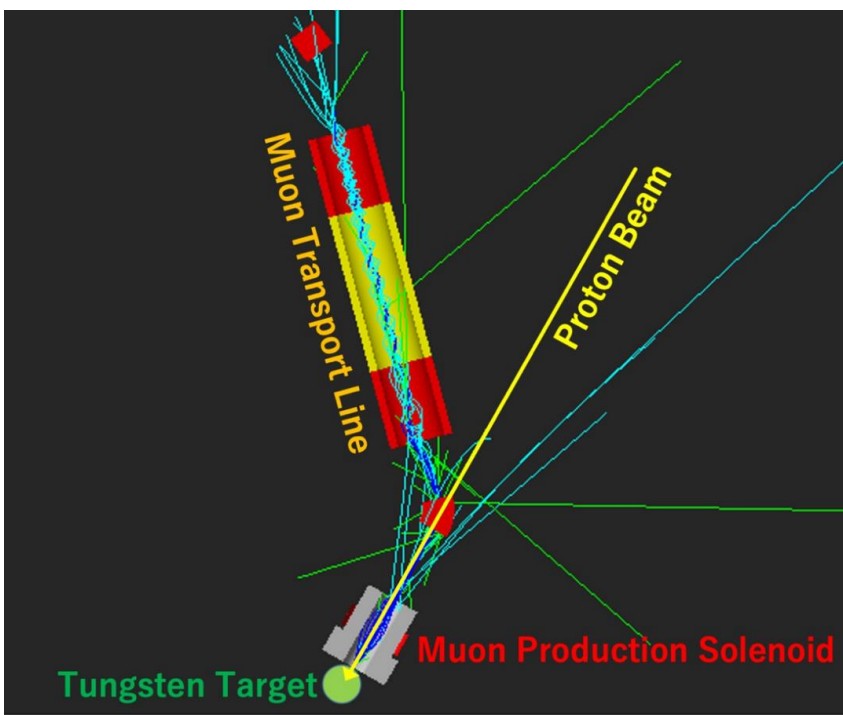

**Figure 6.** Proposed muon beam line for J-PARC MLF second target station TS2.

Figure 7 presents the production solenoid design. The main parameters of the production solenoid are summarized in Table 4. A thick tungsten radiation shield is positioned between the HTS coil and the target to reduce the radiation. The heat deposit density to the coil is a maximum of approximately 40 mW/kg, and the overall heat deposit is approximately 1 kW. A relatively moderate heat deposit density may be achieved using a NbTi magnet. However, with the NbTi magnet, a 4-K helium refrigerator with a high refrigeration power is required. In contrast, with the HTS magnet,

the helium refrigerator that produces a higher refrigeration power at 20 K is utilized for cooling both the production solenoid and neutron moderator. A 20-K multi-kW refrigerator is required for the neutron moderator in any case, and a moderate upgrade of the refrigerator can provide a sufficiently large refrigeration power to cool the production solenoid. This may reduce the overall construction cost sufficiently to compensate for the additional cost required for the HTS conductor. For the HTS scheme, the combination with a better operation efficiency results in the proposed HTS having an obvious advantage over the LTS option.

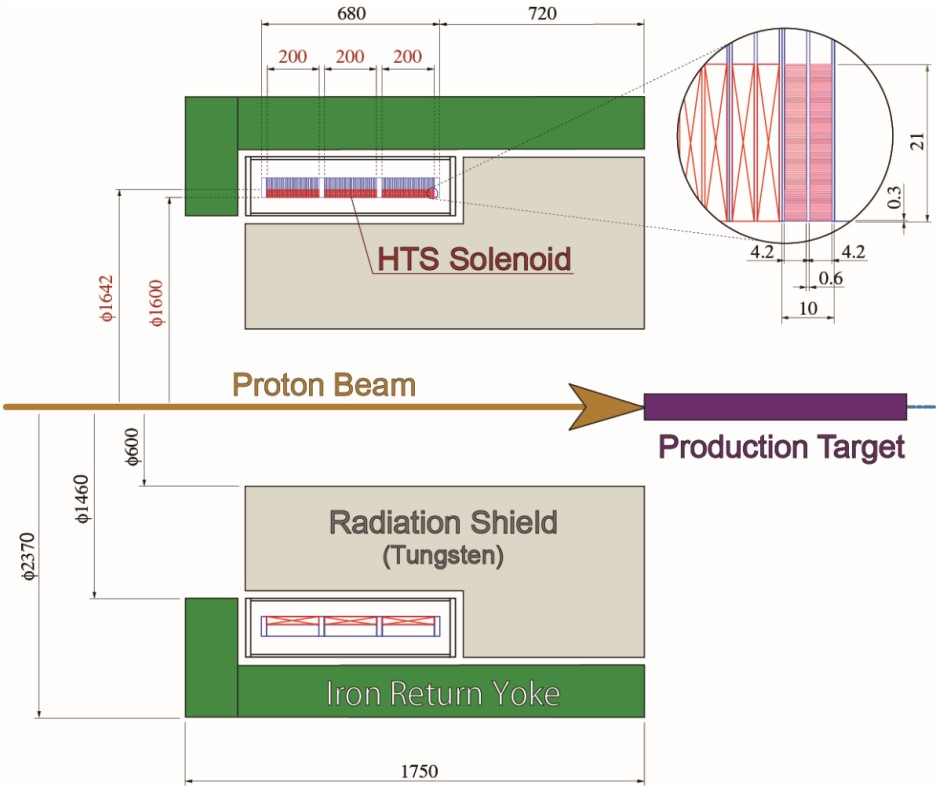

**Figure 7.** Muon production solenoid design J-PARC MLF TS2.

**Table 4.** Main parameters of the muon production solenoid for J-PARC MLF second target TS2.

| Parameter | Value |
|---|---|
| Coil size | ID = 1600 mm, t = 21 mm, L = 600 mm |
| Number of turns per layer | 70 |
| Number of double pancake coils | 60 |
| Transport current | 200 [A] |
| Peak field @solenoid axis | 1.12 [T] |
| Peak field @coil | 2.41 [T] (B//ab: 2.09 [T], B//c: 2.25 [T]) |
| Load line ratio | 48 [%] |
| Peak radial magnetic force of coil | 78 [kN] |
| Axial magnetic force of cold mass | 694 [kN] |
| Peak stress @coil | 38 [MPa] |
| Conductor length per double pancake coil | 733 [m] |
| Total cable length | 44 [km] |
| Conductor cost ($80/m) | 3.52 [M$] |

The annual integral neutron flux is estimated to be about $8 \times 10^{20}$ n/m$^2$/y. Because the neutron irradiation test performed by TU Wien indicated that most of the ReBCO superconductor can withstand a neutron flux of the order of $10^{22}$ n/m$^2$, the solenoid may survive for more than 10 years. However,

the spectrum of the radiation for the solenoid may vary from that of the reactor, and the degradation threshold can be altered. The spectrum may be changed by the shield shape and material combination. A careful design of the shield that avoids an earlier degradation of the ReBCO conductor should be performed. The choice of the ReBCO rare earth material is also a significant factor. The gadolinium-based ReBCO is highly sensitivity to thermal neutrons [25] because Gd has a high neutron cross section of approximately 49,000 barn [34]. Alternative rare-earth elements with lower thermal neutron cross sections, such as yttrium (~8.9 barn [34]) or europium (~4600 barn [34]), have the potential to achieve a lower sensitivity to thermal neutrons. Several studies on irradiation tests for various ReBCO conductors should be performed.

The sol-gel inorganic insulation may be used either directly on ReBCO tape or on metal tape that co-winds with the ReBCO tape conductor. Because quench protection is a sensitive issue for HTS magnets, the effective engineering current density should be reduced to acquire a sufficient duration to shut down the operation current of the magnet. One solution may be the use of copper or aluminum tape for the co-wind tape material.

The use of inorganic materials for electric insulation provides favorable thermal conductivity within the coil structure, thus deeming the conduction cooling structure with helium gas piping attached to the coil outer surface a suitable design candidate. The multi-kW refrigerator can supply adequate gas flow to cool the coil under the operating temperature with a comfortable margin.

The materials used for the cryostat should be radiation-hard. Multi-layer insulation (MLI) that is used for vacuum insulation generally comprises an organic film with an aluminum coating. Alternative technologies with inorganic materials or compromising system design may be required.

A maintenance scenario for the magnet system is also an important issue. Because the system will be highly radioactive after long operation, care is required during remote handling and de-commissioning design. The designs of LTS magnets such as the FRIB fragment separator [20] can be considered as references.

## 5. Discussion

As described in the previous chapter, HTS magnet technology can be a satisfactory technological solution for muon production solenoid magnets with high-intensity muon beam lines. The study indicates that depending on the muon production source and the construction of the radiation shield, the neutron irradiation level can be large even with a moderate heat load to the coil. In this case, degradation of the HTS conductor may be a limiting factor in the system design. Although the heat load is moderate enough to realize the design with LTS conductors, there is an obvious advantage with HTS conductor technologies that streamline the cryogenic part of the design.

The technology also provides a good solution for compact muon sources, as shown in Section 3.1. In the case of a compact muon source, a cryo-cooler should be used for the refrigerator. Because the cryo-head of the cryo-cooler should be attached directly to the magnet, the radiation hardness of the cryo-cooler is also a concern. For Gifford-McMahon (GM) cryo-coolers, Teflon like material is inevitable in the cryo-head because it contains a moving regenerator inside the cryo-head, thus limiting the radiation hardness of the GM cryocooler. The pulse tube cryocooler may be converted to more radiation-hard construction. Because any moving part can be avoided in the cryo-head of the pulse tube cryo-cooler, radiation hard materials can be chosen in the cryo-head.

## 6. Conclusions

This paper discussed the potential for the use of HTS magnets in muon production solenoids. A technological review required for HTS magnets used in a high-radiation environment is discussed. The ability of HTS conductors to enable higher temperature operation yields a better operational margin under high heat load operation conditions due to irradiation. However, hadronic irradiation of the superconductor eventually leads to the degradation of its superconductivity. The typical limit is approximately $10^{22}$ n/m$^2$. A careful study of the overall system and maintenance scenario is required.

**Author Contributions:** Conceptualization, T.O., M.I., N.K., and M.Y.; formal analysis, M.I. and N.K.; writing—original draft preparation, T.O.; writing—review and editing, T.O., M.I., N.K., and M.Y. All authors have read and agreed to the published version of the manuscript.

**Funding:** This research was supported by the Japan Society for the Promotion of Science (grant number 18KK0087).

**Acknowledgments:** The authors would like to thank the members of the Muon Science Section, MLF Div., J-PARC Center; Cryogenics Section, J-PARC Center; and Cryogenics Science Center, KEK for their support. The authors also would like to thank Editage (www.editage.com) for English language editing.

**Conflicts of Interest:** The authors declare no conflict of interest.

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
