# Peer review of "Development of Radiation-Tolerant HTS Magnet for Muon Production Solenoid"

_instruments, doi:10.3390/instruments4040030_

Round 1

Reviewer 1 Report

The paper reports on the possible application of a High Temperature Superconducting (HTS) solenoid for muon production. The main aspect treated in the paper is radiation hardness. Based on state of the art studies the authors suggest possible solutions of materials to be applied in the solenoid for the insulation. The geometry of the magnet is presented, as well as the expected magnetic field.

I recommend publication after the following changes have been made:

  • The English needs to be improved all over the paper, i.e. sentence at line 26 is completely unclear, as the ones at line 30-31 and 34-35.
  • Acronyms as COMET, J-PARC and names of quantities as ID, t, L must be defined.
  • Some discussion on the magnetic design supporting Table 3, as, must be included.
  • The homogeneity region of the solenoid shall also be specified. Ift should also be shown if the specifications are reached with the presented design.
  • Please add a sketch of the coil in section 3.1 to help the reader to follow the text.
  • The captions of figure 5 and 6 are named ‘Figure 10’. This shall be corrected.
  • The numbers in the drawing of fig 6 must be made readable.
  • The authors show in table 3 that 44 km of HTS cable are needed to produce such an HTS solenoid. Is this correct? Since, at least to my knowledge, presently only few m of high quality HTS tape are available commercially, a comment on the availability or plans for production of such a length of material shall be made.

Please in your reply indicate all changes made in the text referring to the parts of text changed.

Author Response

The paper reports on the possible application of a High Temperature Superconducting (HTS) solenoid for muon production. The main aspect treated in the paper is radiation hardness. Based on state of the art studies the authors suggest possible solutions of materials to be applied in the solenoid for the insulation. The geometry of the magnet is presented, as well as the expected magnetic field.

>Thanks for the comments, here is the answer from the authors,

I recommend publication after the following changes have been made:

  • The English needs to be improved all over the paper.
    • English was edited by an English editing service.
  • Acronyms as COMET, J-PARC and names of quantities as ID, t, L must be defined.
    • Corrected
  • Some discussion on the magnetic design supporting Table 3, as, must be included.
    • Added on page9, marked with yellow.
  • The homogeneity region of the solenoid shall also be specified. Ift should also be shown if the specifications are reached with the presented design.
    • No homogeneous region, design are optimizeddirectly with beam simulation code (try to derive the best efficiency reasonably achieved) and beam line characteristic is specified with the simulation results.
  • Please add a sketch of the coil in section 3.1 to help the reader to follow the text.
    • added
  • The captions of figure 5 and 6 are named ‘Figure 10’. This shall be corrected.
    • corrected
  • The numbers in the drawing of fig 6 must be made readable.
    • The figure ischanged and texts are now readable.
  • The authors show in table 3 that 44 km of HTS cable are needed to produce such an HTS solenoid. Is this correct? Since, at least to my knowledge, presently only few m of high quality HTS tape are available commercially, a comment on the availability or plans for production of such a length of material shall be made.
    • Presently the ReBCO conductors of few hundredmeters are commercially available from companies like Superpower or Fujikura, with the price around a  few tens of dollars  per meter.  Some sentences (page 2 yellow part) and Reference 7 is added.

Reviewer 2 Report

The manuscript by T. Ogitsu et al. reviews the characteristics of LTS and HTS magnets showing their applicability for muon production solenoid. In particular, they underline how the HTS could work better than LTS materials.

The topic of the manuscript is interesting but it has to be better written. Therefore, It could be considered for the publication in MDPI Instruments journal after some marks have been clarified.

Specifically:

  • In Figure 1 several components of the COMET production solenoid are shown but it is not clear where are Al strips are as well as what is the path that the He flow does.
  • The authors explain that a solenoid can be made by Low critical Temperature Superconductors such as Aluminum. Since bulk Aluminum has a critical temperature around Tc=1.2K and, when it is in the superconducting phase, does not show a good thermal conductor behavior. Could the authors better specify if the solenoid is cooled down the Al Tc such that the used Al is in its superconducting phase? In the former case, could the authors specify how the heat is dissipated through the solenoid? Instead, if the Al works like a normal material what is the advantage to use it instead of a normal material?
  • The authors make a computation thermal model and compute the heat load distribution. Could the they specify if in their Monte Carlo code both normal and superconducting properties of material have been taken in account?
  • The authors indicate that YBCO and EuBCO could work better than other materials. Could they argue more that sentence even including some reference? 

Few minor remarks:

  • Despite the fact that in the text they refer to figure 5 and 6, nor figure 5 neither figure 6 exist.
  • The authors indicate GdBCO compound properties but no any reference has been indicate. For a better scientific understanding of material properties the authors should add the few main references regarding that material.
  • The manuscript shows several English mistakes as well as a wrong figure number list and some incorrect unit of measure abbreviation ( see 36th line: WM-> MW);

Author Response

The manuscript by T. Ogitsu et al. reviews the characteristics of LTS and HTS magnets showing their applicability for muon production solenoid. In particular, they underline how the HTS could work better than LTS materials.

  • Thanks for the comments, here are the authors answers,

The topic of the manuscript is interesting but it has to be better written. Therefore, It could be considered for the publication in MDPI Instruments journal after some marks have been clarified.

>English is corrected by an English editing service.

Specifically:

  • In Figure 1 several components of the COMET production solenoid are shown but it is not clear where are Al strips are as well as what is the path that the He flow does.
  • > Details of the coil structure is shown in Figure 2, the mark is added in Figure 1 to show Figure 2 part.
  • The authors explain that a solenoid can be made by Low critical Temperature Superconductors such as Aluminum. Since bulk Aluminum has a critical temperature around Tc=1.2K and, when it is in the superconducting phase, does not show a good thermal conductor behavior. Could the authors better specify if the solenoid is cooled down the Al Tc such that the used Al is in its superconducting phase? In the former case, could the authors specify how the heat is dissipated through the solenoid? Instead, if the Al works like a normal material what is the advantage to use it instead of a normal material?
  • > The superconductor is NbTi, section 1 and 3.2 is corrected to clarify.
  • The authors make a computation thermal model and compute the heat load distribution. Could the they specify if in their Monte Carlo code both normal and superconducting properties of material have been taken in account?
  • > The thermal model is independent code to Monte Carlo (PHITs) and it takes into account the material properties. PHITs is used to derive input parameter (heat load) to the thermal model code. In page 5 yellow part texts are corrected to clarify it.
  • The authors indicate that YBCO and EuBCO could work better than other materials. Could they argue more that sentence even including some reference? 
  • > Page 11, yellow part texts are added with reference 31

Few minor remarks:

  • Despite the fact that in the text they refer to figure 5 and 6, nor figure 5 neither figure 6 exist.
  • > Corrected.
  • The authors indicate GdBCO compound properties but no any reference has been indicate. For a better scientific understanding of material properties the authors should add the few main references regarding that material.
  • >Reference 7 is added.
  • The manuscript shows several English mistakes as well as a wrong figure number list and some incorrect unit of measure abbreviation ( see 36th line: WM-> MW);
  • >Corrected

Round 2

Reviewer 2 Report

Since the Authors of the manuscript “Development of Radiation-Tolerant HTS Magnet for Muon Production Solenoid” have clarified the pending questions underlined in the last manuscript version by adding more extensive information in it, no more questions are addressed to them.